# Content-Aware Proactive VR Video Caching for Cache-Enabled AP over Edge Networks

Jinjia Ruan [1,2] and Dongliang Xie [2,*]

1  China Waterborne Transport Research Institute, Beijing 100088, China
2  State Key Laboratory of Networking and Switching Technology, Beijing University of Posts and Telecommunications, Beijing 100876, China
*  Correspondence: xiedl@bupt.edu.cn

**Abstract:** With the rapid development of virtual reality (VR) video networked applications, the use of network caching mechanisms to guarantee the quality of VR services has been proven to be a very effective method. Most of the existing methods on cache placement prediction only consider the one-sided information of user viewpoints and do not consider the video characteristic information of virtual reality, because the asymmetry of the two types of information causes the accuracy of current predictions to gradually decrease, which affects the cache hit rate and leads to VR performance metrics that cannot be guaranteed. In this paper, we analyze the demanding requirements of VR for low latency and high bandwidth in a multi-access point (multi-AP) scenario environment, and further improve the cache hit rate of user requests by increasing network throughput. First, the throughput of VR users after associating APs is analyzed using a Markov model. Second, a nonlinear mixed integer programming problem is constructed with the goal of maximizing the overall throughput of the network system. Finally, combining the characteristics of the VR video content itself and the popularity of the requested video content, the symmetry of the information is guaranteed by considering the ratio between the video characteristic information and the user feature information to determine the weights. The experimental results demonstrate that the proposed algorithm achieves the improvement of cache hit rate and the improvement of network throughput while ensuring the quality of service.

**Keywords:** VR; content-aware; cache-enabled; edge networks

## 1. Introduction

With the continuous development and advancement of VR technology's interactivity, the demand for VR service quality metrics is more stringent than that of ordinary 2D video. The study of Huawei white paper [1] points out that the current stage of VR video should ensure both the basic video quality requirements and the bandwidth of at least 100 Mbit/s in the case of part of the entry-level VR level of weak interaction, and how to guarantee the quality of VR video viewing while considering both the fluency of users in the process of switching viewpoint interaction is a research-worthy issue. Therefore, reasonable caching at the network edge is needed to alleviate the user's dilemma in the face of various unsatisfactory network conditions.

Some existing research approaches on web caching for VR services focus on reducing network latency and bandwidth consumption by deploying proxy caches between users and content servers, thereby reducing the need to pull needed content closer. Study [2] designs an optimisation framework that allows base stations to choose cooperative caching/streaming/edge computing strategies that allow them to maximise the total payoff in serving users, for a given cache/computing resource per base station. The study [3] proposes a new framework that uses cellular-connected drones to collect VR content for wireless delivery. In this model, UAVs can transmit VR content to ground-based

SBSs via wireless backhaul links. meanwhile, SBSs can decide whether to request and store visible or 360-degree video content to reduce backhaul traffic. The study [4] minimises computational latency by exploiting information about the user's pose. By exploiting information about the user's pose, active computation and caching are used to pre-compute and store the user's HD video frames.

Based on the summary of the above studies and related research, the current research faces the following two main challenges. For point-of-view-based prediction, most of the research is based on the prediction of changes in head motion trajectories, but the accuracy decreases significantly over time, and the accuracy of video content caching is also affected. Although VR content-aware caching strategies based on VR content have many advantages, they also face significant challenges. The solution to the tile-based 360-degree video streaming problem is typically to cache all tiles of video for the entire 360-degree video at all resolutions, regardless of whether it is currently in FoV. Thus, tiles of appropriate resolution need to be used to fill the cache of limited size. The study [5] caches video tiles at the highest resolution only and applies transcoding methods for lower resolutions. However, this approach imposes excessive processing requirements on the cache server, which may lead to a waste of computational resources and also a certain amount of time. Therefore, timely cache content switching is needed to improve the efficiency of cache management and reduce the waste of cache space due to the lack of user interest in the viewed video content. To better address the problems faced by the current study, we propose a perceptual approach that takes into account the characteristics of VR content.

For VR latency requirements, more accurate prediction methods and caching mechanisms for content-aware viewpoints are an urgent need to provide higher quality VR experiences. When the AP caching capability is enhanced, along with the improvement of cache hit rate, it can better reduce the user latency and improve the network throughput, so the appropriate AP caching strategy has an impact on the improvement of VR service quality. In this paper, we combine user content popularity with content salience based on VR content request popularity, and we propose the concept of content weighting based on the idea of user psychology behavioral retinal effect [6] (people will be interested in the desired object and will naturally pay attention to the relevant information), based on the interest popularity of the content viewed by user users and the human visual attention mechanism based on it. We propose the concept of content weighting to determine the AP's placement strategy and replacement strategy for content caching, to optimise the overall performance such as network throughput, and to improve the quality of user experience. We give the heuristic algorithm and, finally, verify the algorithm by simulation.

The main original contribution of this paper is in the joint perception of VR video clip popularity and VR video saliency for video edge network caching strategies. Edge cache optimisation research is conducted for the throughput maximisation problem of VR video services under the framework.

1. First, the characteristics of the framework and its cache structure are analyzed, a system model based on VR user cache placement is designed, and cache management is designed and analyzed.
2. Then, the user throughput is analyzed based on a bidirectional Markov model and the problem of maximizing the quantised system throughput is given. The active edge caching strategies of "content popularity awareness" and "joint content popularity and video saliency awareness" are proposed to guarantee the symmetry of information, and the specific flow of the algorithm is given.
3. Finally, the performance of the proposed caching algorithm is evaluated in simulation, and the proposed approach effectively improves the throughput of the entire network and also improves the hit rate of VR caching.

The structure of this paper is as follows. Section 2 presents the related research work. Section 3 constructs the VR user network model as well as the cache placement model. Section 4 details the construction of the throughput analysis model. Section 5 unites content-

aware methods for content popularity and content salience, and constructs a content-aware caching policy based on content awareness. Section 6 validates the performance of the proposed scheme through simulation. Section 7 concludes the paper.

## 2. Related Work

### 2.1. Quality Assurance Strategy Based on Cache Enablement

Caching will have some impact on VR performance. One study considered optimizing the parameters of individual base station slow storage [7], and other studies investigated hierarchical caching in cellular backhaul networks [8,9]. In this study, the information theory of hierarchical caching was investigated, which runs simultaneously on client devices (personalised view cache), the edge, and the cloud, which may require novel multilevel caching architectures [10,11]. In particular, when caching is pushed to the edge, the traditionally understood approach of caching massive amounts of data may no longer be applicable. In addition, personalisation and viewport-driven strategies should be investigated instead of traditional caching approaches to capture the spatial and temporal localisation induced by the user's navigation of VR data. Similarly, it is important to understand how the interaction of virtual and physical features in such applications affects caching, which is another new source of expected data localisation that can be exploited. Several issues related to caching in VR systems have been investigated [2–4,12]. In some studies, existing caching techniques have been used to exploit various cross-sectional information, such as user location, personalisation characteristics, movement patterns, and social relationship attributes, to determine what content to cache and where to cache it, improving the efficiency of accessing content servers on demand.

### 2.2. Content-Aware Optimisation-Based Approach

Most content-based prediction algorithms use significant detection and neural networks to understand the region of interest (ROI) of VR content. Predicting ROI for 360-degree video is inherently different and more challenging than the traditional 360-degree video, because the 360-degree video is omnidirectional. It cannot meet the requirements of real-time video streaming. Therefore, we delve into the viewing behavior of different users to understand the video content. There are two main solutions. (1) to determine the future view area by exploiting the strong correlation of the user's viewing content. Borji et al. studied the content-related features of still images and the prediction of important target detection [13]. (2) Another class of methods is the prediction by highlighting features of videos. Advanced machine learning techniques are often used, through the adoption of various supervised learning methods, including neural networks, for better feature extraction and prediction accuracy in gaze detection [14,15]. We believe that it is intuitive to measure the user's head motion (i.e., viewport) and prefetch the tiles that the user will use. However, many challenges remain in designing such a system, the first of which is a high degree of accountability. We should be highly responsive to fast-paced viewport changes and viewport prediction updates. Secondly, the processing capability must be reasonable. We need to design where the predictions will be executed and may need to define the processing capacity of the device. Finally, there is a need to match temporal changes. The time window for viewport prediction accuracy limits the total time budget for the entire process.

## 3. System Model

### 3.1. Network Model

We consider a scenario, as shown in Figure 1, with cache-enabled APs deployed in close proximity to customers. the AP has limited cache capacity, and it needs to make room when the cache capacity is full. The cache replacement policy may have a significant impact on the utility of the system. In the previous chapter, we only considered the impact of some short-term caching on the system at a short, local scale. We use an access controller to sense and manage the content of user requests. In this architecture, we consider the

global user cache. The following two cases may occur: cache hits and cache misses. If the buffer has the requested content, it sends back the content immediately. Otherwise, it first downloads the requested content from the remote VR video server and then sends it back to the client. Cache misses increase the download latency of the client and also increase the bandwidth consumption. Thus, increasing the cache hit rate brings gain to VR service quality assurance.

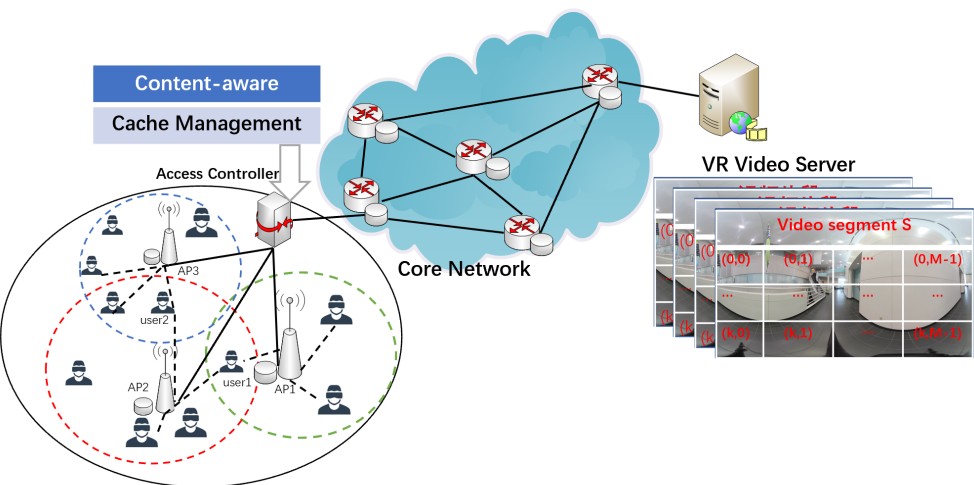

**Figure 1.** System model.

The set of 360-degree VR videos is stored in the VR video source server, which we denote by $F = \{1, 2, \ldots f, \ldots, |F|\}$, and any 360-degree VR video f can be divided into S video segments, denoted as $\mathcal{P} = \left\{f_1, f_2, \ldots, f_{|S|}\right\}$. Each video clip $f_s$ can be further split into $|\mathcal{X}| \times |\mathcal{M}|$ view blocks and the set is defined as $\{\mathcal{X}, \mathcal{M}\}$, and any view field can be represented by $f_s(x, m)$, where $x \in \mathcal{X}, m \in \mathcal{M}$.

With $f_s(x, m)$ as the reference, the adjacent left and right view blocks can be represented as $f_s(x - 1, m)$, $f_s(x + 1, m)$, upper and lower view blocks $f_s(x, m - 1)$, $f_s(x, m + 1)$. $\Psi = \{f_s(x, m) \mid f \in \mathcal{F}, s \in \mathcal{S}, x \in \mathcal{X}, m \in \mathcal{M}\}$ is denoted as the set of all fields of view that can be requested by the user. Moreover, considering that VR videos have different resolutions and different versions, the size of $f_s(x, m)$ can be represented by $SIZE_{f_s(x,m)}$. In order to guarantee the storage capacity of the current network space for the cached content, it is agreed that the current content sum is smaller than the current network cache capacity.

### 3.2. Cache Placement Model

In the early stage of caching, due to the lack of user data, we cannot complete the perception of user content, and can only pre-cache the content that AP-connected users have recently watched with high frequency at the AP, which we call the initial stage, and obtain the initial caching status and user association status by relying on the perception of users' daily viewing behavior habits. For the traditional caching strategy, usually by analyzing the content requested by the user at the AP, it helps to anticipate the content that the user may need in the next phase, and caching this part of the content in advance helps to improve the overall cache hit rate of the AP. The content requested by different users varies greatly, especially for VR 360-degree videos, where even if the same content is requested, the requested viewports are very different. This variation in the popularity of the requested content makes it more difficult for AP to cache VR content. Although the popularity of user-requested content is time-varying, it still has a certain distribution pattern in the short term, such as the typical Zipf distribution. By identifying a certain continuity of interest in user requests over a short period of time, this suggests that the popularity of user-requested content over a short period of time can be inferred from this. Over time, VR video blocks need to be adjusted with the popularity of content requested by VR users, and typical

cache replacement algorithms use Time Aware Least Recently Used, TLRU [16]; with Least Frequent Recently Used, LFRU [17] ), none of the algorithms in this category consider the characteristics of the video content requested by the user of VR. Therefore, we consider the VR 360-degree video, due to the vast view of each frame, and the analysis of video content is also increasingly concerned, assuming that the content salience information of the required video can be obtained in advance by the video detection module, and the higher the probability of the video area being viewed by the retinal effect, the more this behavior is not affected by the popularity of the video. After a period of accumulation of user data information, the AC will accumulate a large amount of user and video content data, through the perception of VR user content, from which we can obtain the user's demand for content information and then cache content placement, and then adjust the AP cache based on the placement of content; we call this stage the cache adjustment period.

This section considers the edge network scenario of dense AP deployment with cache enablement, where an AC is controlled by $i$ APs and $j$ users, and where the AC has the function of cache management and content awareness, and connects the AP to the core network, and the AP controls the user access to the network, and also caches the requested content in advance, based on the results of AC awareness. This chapter assumes that the AP has a set $I$ and the cache capacity is the same size $\mathcal{S}$. Assuming that when a VR user needs to initiate a data request, a suitable AP is selected for association through a sensing method based on user behavior, and a request is sent to the associated AP, there is enough cache space in the network to satisfy the current resource cache, and the sum of all resource contents is less than the cache space, $M$. The resource contents are defined as a set $C$ that contains $c$ different contents and resources, with content size $SIZE_{f_s(x,m)}$. $r_{ij}$ representing the data transfer rate between AP $i$ and user $j$. Therefore, in the edge network scenario, user $j$ requests content $k$ from the network according to his preference. We use $q_{jk}$ to denote the request of user $j$ for content viewport content $f_s(x,m)$. $q_{jk}$ is 1 if user $j$ requested content $f_s(x,m)$, otherwise $q_{jk}$ is 0 and $q_{jk} \in \{0,1\}$. In particular, this can be expressed by the following in Equation (1).

$$ca_{ik} = \begin{cases} 0, & AP_i \text{ uncached content } f_s(x,m) \\ 1, & AP_i \text{ cache content } f_s(x,m) \end{cases} \quad i \in I, k \in K \tag{1}$$

A user can only select one AP to associate with at a given moment, and the association between the user and the AP can be expressed specifically using the following in Equation (2).

$$A_{ij} = \left\{ \begin{array}{l} 0, \quad AP_i \text{ and } User_j \text{ not associated } 1, \quad AP_i \text{ and } User_j \text{ associated} \end{array} \right. \quad i \in I, j \in J \tag{2}$$

$ca_{ik}$ to represent the $AP_i$'s caching of viewport content $f_s(x,m)$. If the $AP_i$ caches content $f_s(x,m)$, then $ca_{ik}$ is 1, otherwise $ca_{ik}$ is 0, and $ca_{ik} \in \{0,1\}$. In particular, it can be expressed by the following Equation (3).

$$ca_{ik} = \begin{cases} 0, & AP_i \text{ uncached content } f_s(x,m) \\ 1, & AP_i \text{ cache content } f_s(x,m) \end{cases} \quad i \in I, k \in K \tag{3}$$

Each $AP$ has a fixed size of cache space $S$. The total size of its stored content cannot exceed the size of the overall cache space under the current network, as expressed by the following in Equation (4).

$$\sum_{k}^{K} ca_{ik} \cdot SIZE_{f_s(x,m)} < \mathcal{S} \forall i \in I \tag{4}$$

## 4. Problem Modeling

### 4.1. Throughput Analysis

According to the network scenario as an edge network with a dense deployment of APs, multiple users associated with the same AP is a concern; when multiple users associated on the same AP will use the same channel for information transmission, so there will be interference among the connected users under that AP. To solve the interference problem of multiple users associated under the same AP, this section is inspired by the study [18], which plans to use Markov model to solve it. In the edge network scenario, when the user sends data before the first channel detection if the channel is idle and needs to enter the random evasion waiting phase. The evasion waiting phase has a total of s evasion states, that is, the evasion waiting process can be carried out at most s times; $W_0$ represents the user in the initial evasion state, the evasion window is the minimum value; $W_i$ represents the ith evasion state, and the size of the evasion window is the size of $W_i$ in The last evasion waiting state is the sth time, and the maximum evasion competition window in this state is $W_s$, and then the evasion competition window size is as follows:

$$W_i = 2^i \cdot W_0, i \in (0, s] \tag{5}$$

We use the Markov model to analyze the process: the user evasion state is 0 at the beginning stage; if the data in this state fails to send successfully, the state is automatically added 1 and the user evasion window is doubled; if it has been unable to send successfully, the above state is cycled until the s-1st state, and the current data is discarded; when the evasion state is 0, the next data transmission starts, and if the user successfully sends the data in this process, the evasion state becomes 0. As shown in Figure 2 for the sth evasion state, $p_j$ is used to represent the case when the conflict collision and transmission error cause user $j$ to fail to transmit data, $s(t)$ is used to represent the random process of the evasion phase of user $j$ at time $t$, $b(t)$ is used to represent the random process of the evasion counter of user $j$ at time $t$, and $\tau_{m',n'}$ is used to represent the random process of the evasion counter of user $j$ at $s(t)$. The state is $m'$ and the stable probability that the $b(t)$ state is $n'$.

$$\tau_{m',n'} = \lim_{t \to \infty} P\{s(t) = m', b(t) = n'\}, m' \in [0, s], n' \in [0, W_m - 1] \tag{6}$$

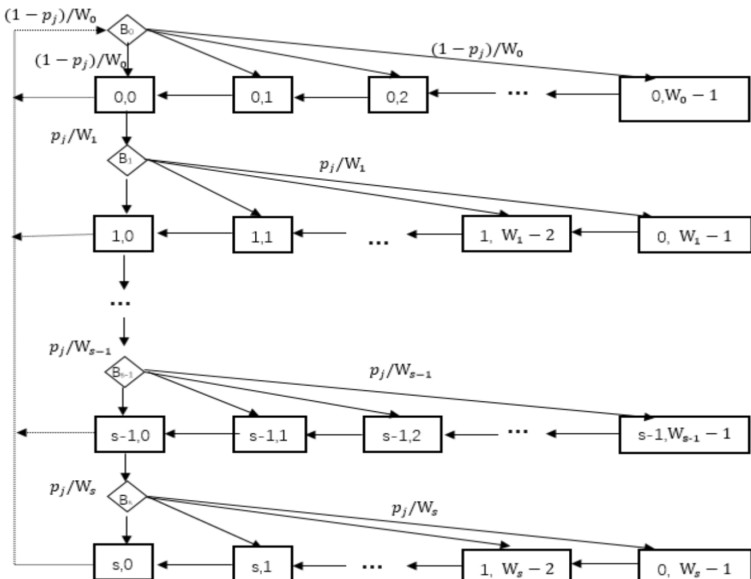

**Figure 2.** Collision avoidance process.

The state transfer matrix of the Markov chain can be represented by Equations (8):

$$
\begin{cases}
P\{m,n \mid m,n+1\} = 1 & n \in (0, W_i - 2) \quad m \in (0,s) \\
P\{0,n \mid m,0\} = (1 - p_j)/W_0 & n \in (0, W_m - 1) \quad m \in (0,s) \\
P\{m,n \mid m-1,0\} = p_j/W_m & n \in (0, W_m - 1) \quad m \in (0,s) \\
P\{s,n \mid s,0\} = p_j/W_s & n \in (0, W_m - 1) \quad m \in (0,s)
\end{cases}
\tag{7}
$$

At the same time, the sum of the probabilities of lying in all states in the Markov chain is 1, that is:

$$
\sum_{m=0}^{s} \sum_{m=0}^{W_m - 1} \tau_{m,n} = 1
\tag{8}
$$

Then, from the above equation, we know that:

$$
\tau_{0,0}(j) = \frac{2(1 - p_j)(1 - 2p_j)}{(W_0 + 1)(1 - 2p_j) + p_j \cdot W_0 \left(1 - (2p_j)^s\right)}
\tag{9}
$$

If we want to obtain the probability that user $j$ can transmit data during the evasive waiting phase, we need to consider the probability that all evasive counters in s evasive states can be reduced to 0, denoted by $\varphi_j$ as:

$$
\varphi_j = \sum_{m=0}^{s} \tau_{m,0}(j) = \frac{2(1 - 2p_j)}{(1 - 2p_j)(W_0 + 1) + pW_0 \left(1 - (2p_j)^S\right)}
\tag{10}
$$

When multiple users of the same AP are associated and there is and only one user $j_1$ for data transmission, the probability that no collision occurs for user $j_1$ data is represented by $ps(j_1)$.

$$
p_s(j_1) = \varphi_j \cdot \prod_{j_1 \in U, j_1 \neq j} (1 - \varphi_j)
\tag{11}
$$

Similarly, the probability $p_f(j_1)$ of sending a collision with other users associated with this AP during data transmission by user $j_1$ can be derived as:

$$
p_f(j_1) = \varphi_j - p_s(j_1)
\tag{12}
$$

Assuming that the wireless channel states are the same, the transmission failure cases due to channel errors can be ignored. That is, successful data transmission can be defined in this paper as sending data without collision, and therefore, the probability of successful data transmission when user $j$ is associated with the AP can be expressed as:

$$
p_s(j_1) = p_s(j)
\tag{13}
$$

We adjust the average delay required for user $j$ to transmit data according to the assumptions to obtain the average network delay $T_{avg}(ij)$ when computing the user $j$ associated with AP in this paper, denoted as follows:

$$
T_{avg}(ij) = T_I(i) + T_S(i) + T_C(i)
\tag{14}
$$

where $T_I(i)$ represents the time interval when no user sends data on the AP associated with user $j$, and $\alpha$ is some constant representing the waiting time interval between two adjacent evasive waiting states. The representation equation is as follows:

$$
T_I(i) = \prod_{j \in U} (1 - \varphi_j) \cdot \alpha
\tag{15}
$$

$T_S(i)$ represents the time required to successfully request content $k$ when user $j$ associates AP. $\cdot$ is a constant used to represent the time required by user $j$ for each evasion phase during the transmission of data. It can be expressed by the following equation:

$$T_S(i) = p_s(j) \cdot \left( \beta + \frac{SIZE_{f_s(k,m)}}{R_{ij} * A_{ij}} \right) \tag{16}$$

$T_C(i)$ represents the time required to confirm a collision between user $j$ and AP during data transmission, and can be expressed in Equation (17) as:

$$T_C(i) = p_f(j) \cdot \left( \beta + \frac{SIZE_{f_s(k,m)}}{R_{ij} * A_{ij}} \right) \tag{17}$$

Finally, we can derive the theoretical throughput size when user $j$ and AP are associated based on the actual model built in conjunction with (18) as:

$$x_{ij} = p_s(j) \cdot \frac{A_{ij} \cdot r_{ij} \cdot T_S(i)}{T_{avg}(ij)} \tag{18}$$

*4.2. Problem Formulation*

In wireless edge networks, APs can obtain different network throughputs under different user association states because APs have certain content caching capabilities. How users can maximise the overall network throughput under content cache-based APs can be modelled as a combinatorial optimisation problem with state shifting with the maximisation of the overall network throughput as the main optimisation objective. In this part, based on the content-aware caching of APs, in order to solve for the throughput of users in the association state, the theoretical value of the throughput of users associated with APs in the problem modelling phase is expressed in Equation (19) as:

$$x'_{ij} = \sum_k^K p_s(j) \cdot \frac{r_{ij} \cdot A_{ij} \cdot T_S(i) \cdot SIZE_{f_s(x,m)}}{T_{avg}(ij)} \cdot P_{ki} \cdot q_{jk} \cdot cac_i \tag{19}$$

We obtain the size of the throughput when user $j$ occurs the association state with cached content $k$ as, where $p_s(j)$ represents the transmission success probability; $SIZE_{f_S(x,m)}$ represents the content size; $q_{jk}$ represents the request of user $j$ for content viewport content $f_s(x,m)$; $P_{ki}$ represents the influence of AP $i$ cached content $k$ on user $j$'s choice of the content-aware impact factor of AP; $T_S(i)$ represents the successful data transmission time; $T_{avg}(ij)$ represents the network delay during the association of user $j$ with AP $i$. The expression for the optimisation objective of the overall network throughput is obtained according to the scenario assumptions as:

$$\max \sum_j^J \sum_i^I x'_{ij} \tag{20}$$

Due to the high computational complexity of the optimisation objective, it is difficult to be solved directly using mathematical tools. Therefore, for this problem, a heuristic algorithm solution scheme will be proposed in this chapter, and the details will be described in the next section.

## 5. VR Content-Aware Caching Strategy

*5.1. Active Caching Strategy for Content Popularity*

This section considers the prediction of user content popularity. Users' own characteristics are generally somewhat different, so they can be mainly divided into the following cases: different users may request different content at different moments, and different

users may request different content at the same moment. The popularity of content is not constant, but varies over time. As the content popularity changes, the original content of the AP cache becomes invalid, which increases the waiting time for user requests and reduces the quality of user experience. It is important for both users and the network to perceive user popularity and to cache content as well as replace it based on this acquired popularity. Since users' psychological cognitive habits are a long-term process, they generally maintain the cognitive characteristics of continuous attention and periodic review of interest content. Based on this characteristic, we believe that the interest preference of user attributes will continue to focus on a certain interest content at a certain time, and users will request similar types of content in a certain period of time.

This section analyzes the caching policy based on the persistence and periodicity of user content requests, predicts the popularity of content, and performs content caching. Therefore, the same content may have different popularity levels at different times, and different content may not have the same popularity level at the same time, i.e., the popularity of content may be different at different times. As mentioned earlier, the popularity of the content also affects the caching rules, so the design of the content caching algorithm in this paper is based on the popularity of the content. However, differences in content popularity are a barrier to selecting cached content. Research [19] shows that the results of the Zipf distribution can be used to describe the popularity of the requested content in the network, however, it is unreliable to determine the trend of the requested content based only on the Zipf distribution, in order to improve the accuracy of the popularity. In this section, we distinguish the popularity into two parts: global popularity and local popularity. Such as in most studies, we also use the Zipf distribution to represent the global popularity of the content, and call $PG_k$ the global popularity of content $k$ according to Equations (21) and (22). Then, the calculation formula is shown as follows:

$$PG_k = \frac{1}{k^r H_{c,\gamma}}, k \in \{1, 2, 3 \ldots c\} \tag{21}$$

$$H_{c,\gamma} = \sum_{k=1}^{c} \frac{1}{k^\gamma} \tag{22}$$

The content set is denoted as $C = \{1, 2, 3 \ldots k \ldots c\}$, $c$ is the number of different contents, $H_{c,\gamma}$ denotes the generalised coordination function, and $\gamma$ denotes the feature index, which is a specified constant. For a single AP, the local content popularity is mainly considered, for users' requests for content in a time interval can be obtained by counting the users associated with this AP, i.e., related to the preference characteristics of the users associated with this AP. In this section, we propose a local content popularity matrix to analyze the relationship between users, content and APs, and determine the local popularity of content $PL_{ki}$ by analyzing the proportion of content requests on APs. The specific expression is shown in Equation (23).

$$PL_{ki} = \begin{bmatrix} f_{11} & \cdots & f_{1a} \\ \vdots & f_{ki} & \vdots \\ f_{c1} & \cdots & f_{ca} \end{bmatrix} \tag{23}$$

where $f_{ki}$ in the matrix row represents AP $i$ requests for content $k$. The more requests for content $k$ from different users associated with AP in a statistical time period indicates that the popularity of content $k$ in AP is higher, i.e., the higher the value of $f_{ki}$, the higher the benefit of caching content $k$, and therefore the higher the network throughput.

It is not possible to determine the exact pattern of individual content requests based on the Zipf distribution alone, so we can determine a more realistic user popularity by analyzing global and local popularity. According to the Zipf distribution, the global popularity refers to the overall popularity of content, while the global popularity of individual content is determined by Equations (21) and (22). Also combined with the users' own

viewing characteristics, the consistency and repetition of users' preferences related to the local popularity of content can be analyzed based on the user request history associated with the AP, and the user local content popularity matrix can be calculated by Equation (23), which finally yields the local popularity of each local. The AC can calculate the frequency of each user requesting network content from each AP. In the initial phase of the network due to the temporary lack of data on the history of user requests, the popularity of a content is calculated from the second association cycle of an AP. In an edge network with densely distributed APs, densely and randomly distributed users continuously send requests to the APs. If APs do not cache the content requested by users, they keep sending requests to ACs via backhaul lines, which greatly increases the load on the wired links between APs and ACs, increases the latency of user requests, wastes many resources, and even causes congestion on backhaul lines. However, if valid content is cached in the AP cache to ensure a high hit rate of the AP cache, it not only reduces the backhaul lines and load between the AP and the AC, but also reduces user latency and improves the quality of service for users.

Therefore, according to the popularity of each content, AC calculates the ranking of the local popularity of each access point based on the ranking of the global popularity of content $k$. The impact of content $k$ on the next step of user requests after it is cached in the AP, i.e., Equations (4) and (5), and finally, the access controller AC calculates the network utility of each access point when caching content $k$, ranks it in descending order, and caches it sequentially at the access points until there is not enough cache space to cache any content, provisions the cache according to the popularity degree of the content, and performs the cache. The cache is provisioned according to the popularity of the content and replaced according to the cache replacement Algorithm 1 when the popularity of the content changes.

---

**Algorithm 1** Content cache placement algorithm based on content popularity

---

**Input:** $\{A_{11}, \ldots, A_{ij}\}, \{r_{11}, \ldots, r_{ij}\}$
**Output:** $\{P_{11}, \ldots, P_{ki}\}$
1: Initialisation, $PG_k = 0, PL_{ki} = 0, P_{ki} = 0, \forall i \in I, k \in \mathcal{S}$;
2: Calculate the global prevalence of content $k$ in one cycle $PG_k$
3: **for** $AP\ i$ **do**
4:      **for** $C\ k$ **do**
5:          Calculate the local prevalence of content $k$ $PL_{ki}$
6:      **end for**
7: **end for**
8: **for** $C\ k$ **do**
9:      **for** $AP\ i$ **do**
10:          Calculate the $PL_{ki}$
11:      **end for**
12: **end for**
13: **for** $AP\ i$ **do**
14:      **for** Every content that was requested in the previous cycle and has not been cached in the current cycle **do**
15:          The current content is listed in decreasing order of popularity by $P_{ki}$
16:      **end for**
17: **end for**
18: **if** $\mathcal{S}_i \geq SIZE_{f_s(x,m)}$ **then**
19:      Select the node with the highest popularity and sufficient cache space in the above queue
20:      $\mathcal{S}_i = \mathcal{S}_i - SIZE_{f_c(x,m)}, P_{ki} = 0$
21: **else if** $\mathcal{S}_i < SIZE_{f_s(x,m)}$ **then**
22:      Discard and cache current content
23: **end if**

---

### 5.2. Content-Aware Cache Replacement Strategy

This section focuses on the cache replacement policy, considers the impact of the actual received content on the cache according to the VR characteristics, and improves the overall hit rate of the caching policy by enhancing the perception of the user-requested content. To better quantitatively represent VR content perception, we propose to use the weight parameter $w_{i,j}$ to represent the perception of content. In the past few years, many studies have emerged to collect and analyze the navigation patterns of users viewing VR content. Most of the studies [20–25] used content-related saliency maps as the main result of their analysis, where the saliency map calculates the most likely region of the sphere that the viewer pays attention to based on the viewer's head or eye movements. The study [26,27] focused on the impact of VR video saliency features on VR users, and the feature code rate expectation of VR content Tile is closely related to the user's saliency (Saliency) feature of the content, where the higher the user's attention is the stronger its saliency, and the saliency represents the degree of attraction of the current content to the user. Therefore, we can consider from the content image level that the quality of the Tile in the high saliency area will directly affect the user's experience, which means that for the whole block, the user needs to get the content in the high saliency area to get more visual experience from it and to improve the viewing experience. Figure 3a,b show the prominence heat map of VR 4X8 Tiles video frames and VR 4X8 Tiles video frames, respectively. Generally speaking, areas with strong saliency are more colorful or textures are more conspicuous, such as the rest area and the green wall map area in Figure 3a, while areas with weaker saliency have a single color or texture, such as the white wall part and the transparent window sill part in Figure 3b, and VR users will pay more attention to details in the salient areas when watching such videos.

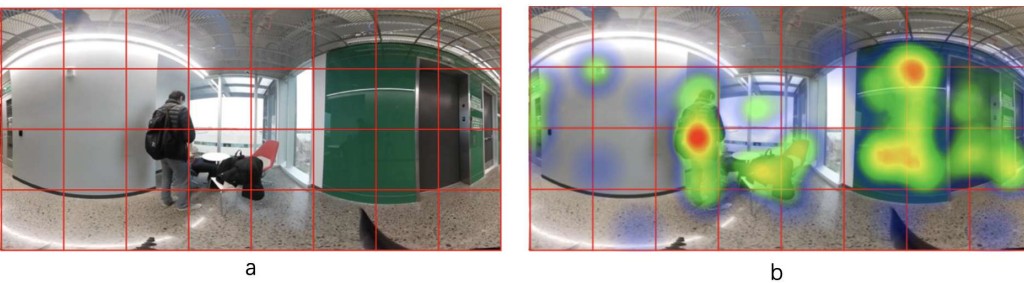

**Figure 3.** (**a**) VR 4X8 Tiles video frames under ERP projection, (**b**) Heat map of VR 4X8 Tiles video frame saliency under ERP projection.

It has been studied [28–30] that by analyzing the existing dataset, it was found that the prediction for the user's viewpoint depends on the historical viewpoint path and image content of its users. As far as image content is concerned, salient content is more likely to attract the viewer's attention. Therefore, the saliency of the content is related to both the appearance and the motion of the objects in the viewed content. Moreover, a general statistical analysis of users' head movements revealed a generally high degree of consistency in spatial distribution among users. Xu et al. evaluated a highly linear correlation between heat maps generated by two random groups of users, with most of the viewing directions falling into small anterior regions located close to the equator [28]. The study in [30] evaluated the mean intersection angle of eye gaze directions for each pair of participants in each VR360 content, an analysis that highlights the heterogeneity of user behavior, in contrast to observations from other studies. Most of the aforementioned studies have focused on changes in content salience over time in terms of behavior (e.g., content salience heatmaps). These metrics are highly informative about where users tend to look. However, they are only partially informative for over time, and we can infer the degree of user instability.

In the following, we will focus on the analytical study based on the popularity of user-requested content, and emphasise the determination of user content weights. Based

on the above analysis, for VR content caching, the differences in content demand brought by the different regions of each Tile content will be considered in the caching strategy due to the different prominence of the content in each Tile, in order to compensate for the missing cache blocks caused by the lack of consideration of video content features based on popularity alone. The lack of consideration of the image content characteristics brings about the problem of missing cache blocks. Here, we consider the Tile block as the block corresponding to the saliency information $S_{i,j}$ can be obtained directly by the image saliency detection module. Therefore, in order to fully utilise the cache space of AP, we synthetically consider the user perception weight of the Tile related to the popularity and saliency of users. We synthesise the popularity and saliency impact on users, as shown in Equation (24).

$$w_{i,j} = \eta S_{i,j} + \theta PL_{ki} \tag{24}$$

$$\eta + \theta = 1, \eta \in (0,1), \theta \in (0,1) \tag{25}$$

The weight occupied by user salience at this point we denote by $\eta$, and the weight occupied by popularity at this point we represent by $\theta$. In this paper, we derive the values of both by experimental simulation. As mentioned earlier, this part of the caching Algorithm 1 in this section is based on the popularity of the content accessed by the user. By caching highly popular content near the user's network edge, we can effectively reduce the user's download latency. To improve the success rate of caching, by analyzing the user's request history and obtaining valuable information from it, we show similar preferences for hot content on the network for different users, but also for certain types of content. We also consider the salience characteristics of VR video content, where users need to obtain more viewing content and details in the salience area. Therefore, this section divides the perception of user content into two parts, user popular content, and user salience content, to realise the perception for user-demanded content and construct the weight coefficient $w_{i,j}$ for requested content. The Algorithm 2 flows as follows:

---

**Algorithm 2** Content-aware content cache placement algorithm

---

**Input:** $\{P_{11}, \ldots, P_{ij}\}, \{A_{11}, \ldots, A_{ij}\}, \{r_{11}, \ldots, r_{ij}\}$
**Output:** $\{P'_{11}, \ldots, P'_{ik}\}$
 1: Initialisation, $A_{ij}, \forall j \in J, i \in I$
 2: **for** *User j* **do**
 3:   **for** *AP i* **do**
 4:     **for** *C k* **do**
 5:       **if** $AP_i$ Cached content $k$ **then**
 6:         Calculate the cache hit rate of content $k$ on $AP_i$ in the last scheduling cycle;
 7:         Compute $P_{ki}$, compute the content parameter perception $w_{i,j}$;
 8:       **end if**
 9:       Calculate user throughput based on user $A_{ij}$;
10:     **end for**
11:     Calculate the throughput when user User $j$ is associated with AP $i$;
12:     Compare with the current maximum throughput, record the maximum throughput
13:     **if** Current Throughput $<$ Historical Maximum Throughput&& $S_i - P_{ki} \geq SIZE_{f_s(k,m)}$ **then**
14:       Select the node $P'_{ij}$ with the highest weight in the above queue and enough cache space
15:       $S_i = S_i - P_{ki} - SIZE_{f_s(k,m)}, \quad P_{ki} = 0$
16:     **else if** $S_i < SIZE_{f_s(k,m)}$ **then**
17:       Discard and cache current content
18:     **end if**
19:   **end for**
20:   Execute Algorithm 1
21: **end for**

---

The ultimate goal of this section is to select the most appropriate cached content for the user to maximise the network throughput and improve the caching success rate. This section is designed to use the Algorithm 1 to determine which content is cached on APs, i.e., the access controller determines which content is cached on each AP based on the popularity of the content. This subsection deals with the update and switching of the cache by the access controller based on the content weight after the AP caches the content. The calculation and scheduling process described above is performed by the CA. As a result, the total placement matrix $P_{ki}$ of the cached content is obtained.

## 6. Experimental Analysiss

### 6.1. Simulation Experiments

This chapter will use MATLAB simulation tools to verify the effectiveness and superiority of the algorithm proposed in the previous section. This section first simulates the scenario of the edge network in Section 3.1, as shown in Figure 4, which mainly shows the network topology diagram composed of densely deployed APs and randomly distributed users.The simulation scenario selected in this section is a square area of $100 \times 100$ m², in which 25 AP nodes are evenly placed for user access, and each AP node has the same coverage area of 500 m², ensuring that at least four APs have overlapping coverage areas within an AP group. The neighboring APs use frequency orthogonal channels for data transmission without neighboring frequency interference, so that the APs using the same frequency are far enough apart to ignore co-channel interference. The random distribution of 200 users in this area ensures that each AP has to serve different users, and most of the users are distributed in the overlapping area between adjacent APs, so the performance of the whole network is influenced by the association relationship between APs and this part of users, which lays the foundation for designing more complex and intelligent association strategies for the scenario.

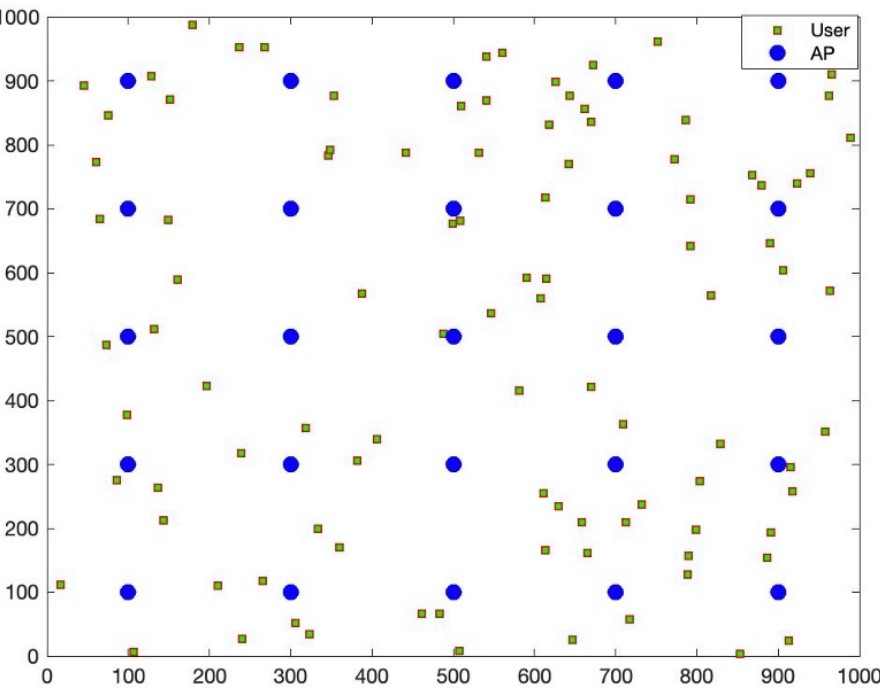

**Figure 4.** Network topology diagram.

Then, the performance of the proposed algorithm in this paper is analyzed in terms of network throughput and AP cache hit rate, respectively. We randomly select the VR video dataset provided by the public dataset [30] for the simulation experiments, considering that the VR video data volume is much larger than the ordinary video, and there are 10

360-degree VR video files in the video library. Each 360-degree VR video file has 100 clips, each clip has 4 × 8 fields of view, and the number of different contents is set to 32,000 in this paper, with equal content size and set to unit 1. The popularity of video files at each place follows the Zipf distribution [31], and we assume that the popularity of full-frame VR videos conforms to Zipf's law with a deviation parameter $\alpha = 0.8$. In the VR video viewing process, the user's viewport switches frequently. In our simulations, we randomly selected 30 users from the public dataset of user behavior [19], whose viewport requests obey the Poisson arrival and departure model with an average arrival time of 3 ms. The number of concurrent active requests is estimated by the M/M/$\infty$ queuing model [32], which follows Little's theorem [33,34] with $N_r = \lambda_r \cdot T_a$, where $T_a$ is the request activity time and $1/\lambda_r$ is the average request inter-arrival time. Requests for 5000 viewports were simulated based on the popularity ranking of VR video tiles.

APs in edge wireless networks all have some caching capability. In this chapter, the cache space size of each AP is set equal to 100, 150, and 200, respectively. In this chapter, we set the duration of the scheduling cycle to 10s. The first phase of the simulation directly uses the correlation results of user behavior perception obtained in the previous chapter, which we call the first phase. Starting from the first stage, the popularity of the content is calculated, analyzed, predicted and cached based on the content requested by the user in the previous stage. Here, we mainly consider the second phase.

### 6.2. Baseline Algorithm

This chapter focuses on the performance comparison of the proposed algorithm with typical baseline algorithms, which is divided into:

- Random caching algorithm: This algorithm performs random placement for cached resources under the assumption that there is no complete prior knowledge of user requests.
- Content popularity caching algorithm: Based on content popularity, it is currently the most commonly used caching algorithm. The algorithm only keeps a sorted list of the number of user requests for content over a period of time to obtain the popularity of user-requested content.
- Content-aware caching algorithm: This algorithm is a comparison algorithm given in this paper. The basic idea is to use big data to analyze user attributes such as the relevance and continuity of user-requested content, and to analyze the degree of preference of content with the salience characteristics of VR videos, and to guide the caching strategy based on the analysis results.

### 6.3. Performance Evaluation

6.3.1. Network Throughput Analysis

Our objective, given in Algorithm 2 regarding user content perception, is the user-perceived content weights described above, since the values of parameters $\eta$ and $\theta$ in $w_{i,j}$ are not determined in Eq. As observed in the Formula (25), the sum of parameters $\eta$ and $\theta$ is 1, and the values of parameters $\theta$ and $\eta$ are decimals between 0 and 1. In the simulation environment, for the convenience of the calculation, and for the selection of the matching value aspect, we mainly refer to the use of the traversal 0.1 to 0.9, since $\eta + \theta = 1$ and the network throughput is affected by the different values of the parameter $\theta$, as observed in the simulation diagram below, when $\eta = 1 - \theta$. We divided 0.1 to 0.9 into two parts in the simulation flow to facilitate the analysis, in order to get the highest throughput $\theta$ value; we then compare 0.1 to 0.5 and then obtain the higher throughput $\theta$ value from it, and then compare that value with the number between 0.6 and 0.9, and then the obtained $\theta$ value. Figure 5a,b are obtained after simulation experiments, as shown in Figure 5a, when $\theta$ is 0.1 to 0.5 and $\theta$ is 0.5 when the network throughput is relatively large. In this knot here, we put 0.5 and 0.6 to 0.9 to conduct a comparison, and thus obtain Figure 5b, when $\theta$ is 0.8, as observed by the figure when the overall network throughput is the maximum. Thus, in the simulation experiment, we set the value size of $\eta$ to 0.2 and the value size of $\theta$ to 0.8.

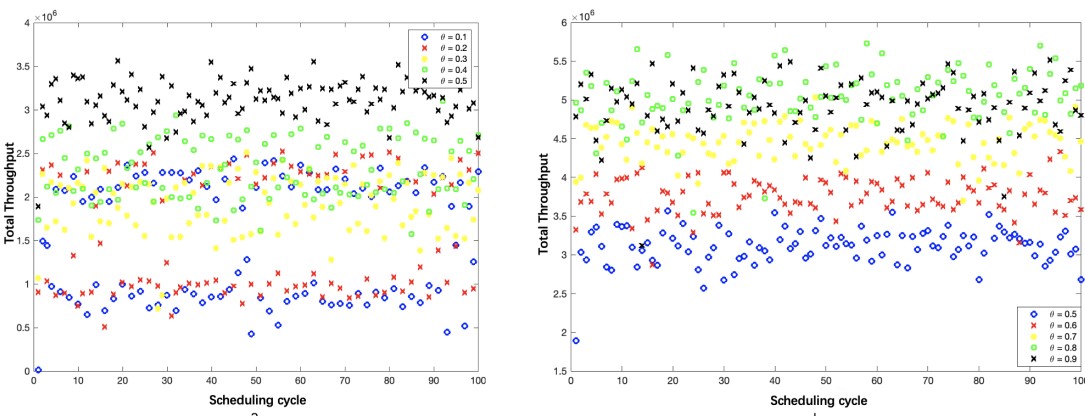

**Figure 5.** (**a**) $\theta$ belongs to 0.1 0.5 throughput; (**b**) $\theta$ belongs to 0.5 0.9 throughput.

We also compare the throughput of different caching schemes over a period of scheduling cycles, as shown in Figure 6. The network throughput of the content-aware caching-based algorithm AP performs better than the popularity-based caching algorithm and the random caching algorithm. For the results, we believe that random caching cannot truly and objectively reflect users' preferences for viewing content, and the user-popularity-based caching strategy takes into account the influence of user preferences to a certain extent but does not sufficiently explore the user-content preferences, especially for VR services with certain interaction capabilities, improving the overall performance of the system. To better compare the advantages and disadvantages of caching strategies, we also analyze the cache hit ratio of the random caching strategy popularity caching strategy and the caching strategies mentioned in this chapter.

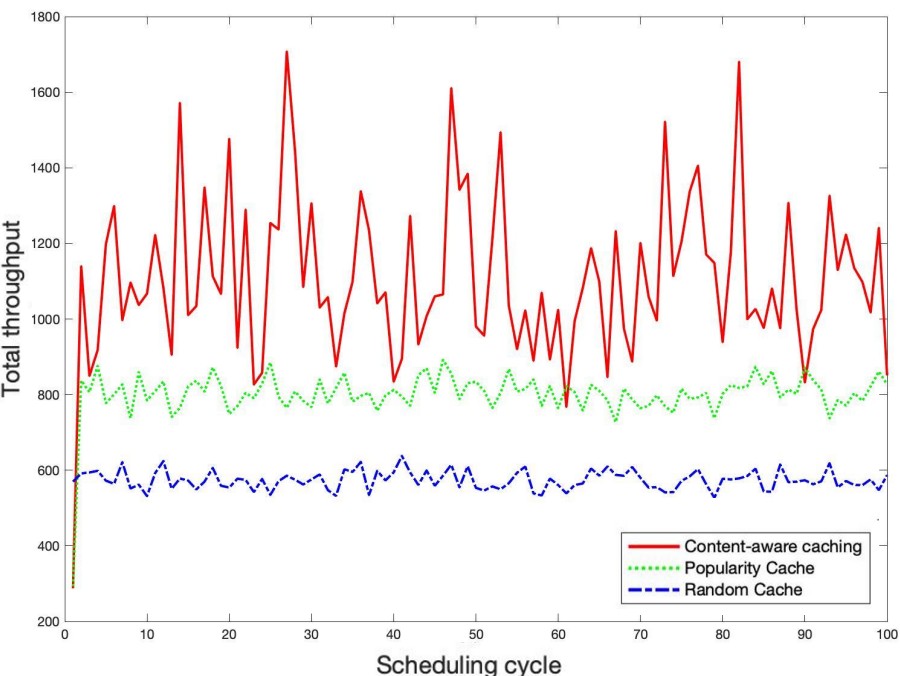

**Figure 6.** Throughput comparison of different caching schemes.

6.3.2. AP Cache Hit Rate Analysis

Cache hit ratio is used as the primary measure of cache performance. It directly reflects the probability that the requested data will reside in the cache. Traditionally, cache hit ratio is calculated based on full-frame video blocks. For the VR video, the basic unit in the caching system is the tile. Therefore, request hit ratio is calculated based on blocks. It is

defined as the number of cache hits divided by the total number of requests for the blocks. For verifying whether the caching algorithm is efficient or not, the cache hit ratio is one of the important performance metrics. We design the caching algorithm for wireless APs, focusing on the consideration of one-hop forwarding between users and wireless APs in wireless edge networks.

We can interpret the cache hit rate as the content of the AP associated with the user that has its own request cached. Here, the simulation verifies the cache hit rate obtained by caching the content popularity and compares it with the cache hit rate when random caching is performed, resulting in the simulation results shown in Figure 7. As shown in the figures, the cache hit rate in the content popularity-based caching policy is improved, indicating that the content popularity-based cache placement algorithm proposed in Section 5.1 of the paper is effective. Figure 7 show the performance of different caching policies in terms of cache hit rate for a given cell implementation, for different cache sizes. It is clear that for all cache sizes, the content-aware caching policy performs significantly better than the other caching policies. For example, when the cache sizes are 100, 150, and 200, the cache hit rates for the content-aware caching policy are 0.67, 0.67, and 0.65, respectively, while the cache hit rates for the popularity caching policy are 0.4, 0.4, and 0.38, respectively. The hit rates for the random cache are all around 0.2.

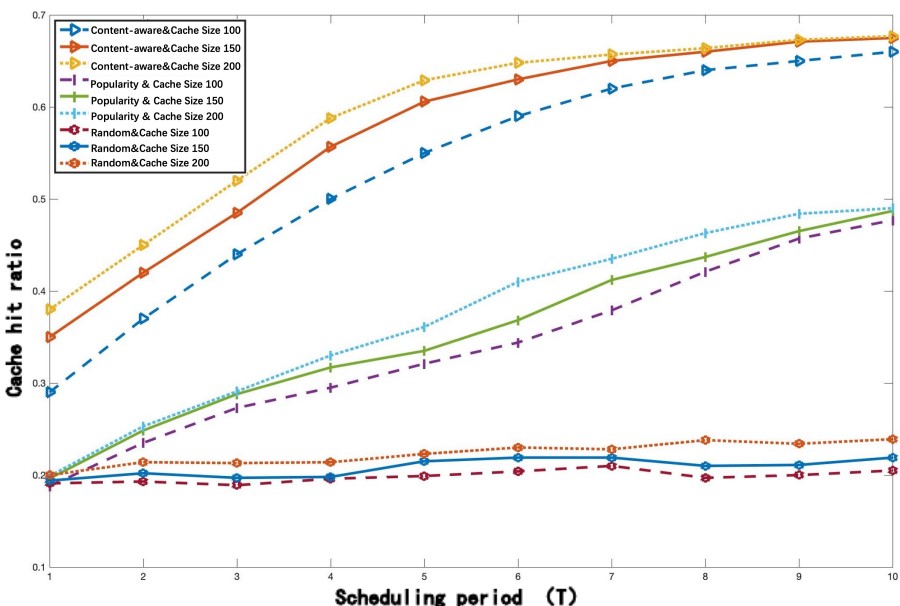

**Figure 7.** Cache hit rate comparison chart.

## 7. Conclusions

In this paper, we propose a pre-caching strategy based on VR user content characteristics, predicting content preferences based on the popularity of user-requested content, and selectively pre-cache the personalised recommended content to the wireless access controller AC for different users. On the one hand, using the large amount of user attribute information obtained in the AC, we analyze the relevance and continuity of user content requests to predict the degree of content preference, and on the other hand, we consider the salience characteristics of VR videos, and establish a pre-caching benefit model based on the results of both analyses to achieve an effective caching strategy. The effective utilisation of cache space and wireless resources is improved, and the user content acquisition throughput is mainly optimised.

Currently, with the high speed of the fifth generation communication technology, multi-access edge computing (MEC) is considered to have a driving role in VR development. Therefore, it is a very interesting research point to study the trade-off between the need to ensure high bandwidth and low latency in MEC-based VR systems and QoE

optimisation. QoE optimisation of VR systems needs to consider the cost of reducing cache and computation while ensuring the user viewing experience. Balancing the QoE of users with the cost of caching and computing resources with the involvement of MEC is a future research direction.

**Author Contributions:** This work was mainly performed by J.R. (planning of the work, conceptualisation, investigation, methodology, data curation, formal analysis, resources, software, visualisation, and original draft preparation) and was completed with key contributions from D.X. (planning of the work, conceptualisation, supervision, validation, manuscript review and editing, and funding acquisition). All authors have read and agreed to the published version of the manuscript.

**Funding:** This research received no external funding.

**Acknowledgments:** I would like to thank my wife (Xiaoyun Ma) and children (Dudu) for their wise advice and compassion. You have always been there for me.

**Conflicts of Interest:** The authors declare no conflict of interest.

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
