# Peer review of "Content-Aware Proactive VR Video Caching for Cache-Enabled AP over Edge Networks"

_electronics, doi:10.3390/electronics11182824_

Round 1
Reviewer 1 Report (New Reviewer)
In this paper, the authors proposed a VR content-aware cache placement strategy that selectively pre-caches personalized recommended content from different users into the wireless access controller AC based on the preference prediction of content popularity. The article is well written and addresses an important issue, but I have a few major concerns about the method and assumptions.
1) Figure 5 and 6 are hardly readable. Please increase the font size in the figure and use a better quality figure.
2) For the throughput analysis. I suggest authors may investigate if it is feasible to adopt the random coding scheme. Because the type of data generated in VR applications usually has a large amount of redundancy. Please refer to this article https://ieeexplore.ieee.org/document/8451876 which adopted the random coding scheme for the distributed caching network like NDN.
3) Please discuss how different caching schemes can impact the proposed scheme, and how the proposed will work for cached content. Please check these two articles to understand how to adopt LFRU or TLRU scheme https://arxiv.org/abs/1702.04078, and https://arxiv.org/abs/1801.00390 . One need to explain the application scenario and then choose the best scheme with proper justification.
4) Why do both algorithm 1 and 2 has the same title “Content cache placement algorithm based on content popularity”
5) Please include a stepwise explanation of the proposed algorithms.
6) Authors stated that “The local popularity can then be calculated based on the user’s own characteristics, combined with the historical request records of users associated with the same AP”. Why do we need to find the local popularity based on a user’s local characteristics? It will be more appropriate to consider the location-based popularity by taking the edge devices into consideration.
Author Response
Response to Reviewer 1 Comments
We thank the reviewer for comments. Our reply is detailed below.
Revised as suggested.
Special thanks to you for your good comments.
In this paper, the authors proposed a VR content-aware cache placement strategy that selectively pre-caches personalized recommended content from different users into the wireless access controller AC based on the preference prediction of content popularity. The article is well written and addresses an important issue, but I have a few major concerns about the method and assumptions.
1) Figure 5 and 6 are hardly readable. Please increase the font size in the figure and use a better quality figure.
Response 1:Thank you very much for your suggestion, we re-increased the font size in the figure again and used a better quality figure.
2) For the throughput analysis. I suggest authors may investigate if it is feasible to adopt the random coding scheme. Because the type of data generated in VR applications usually has a large amount of redundancy. Please refer to this article https://ieeexplore.ieee.org/document/8451876 which adopted the random coding scheme for the distributed caching network like NDN.
Response 2:Thank you very much for your suggestion, I think it is a better solution, our team has done research on this before () and has done similar work, but this time this solution does not seem to be suitable. Still, thank you for your suggestion and we will add it to the discussion.
Wang Q, Xie D, Ji X. Network codes-based content-centric transmission control in VANET[C]//2015 international conference on connected vehicles and expo (ICCVE). IEEE, 2015: 157-162.
3) Please discuss how different caching schemes can impact the proposed scheme, and how the proposed will work for cached content. Please check these two articles to understand how to adopt LFRU or TLRU scheme https://arxiv.org/abs/1702.04078, and https://arxiv.org/abs/1801.00390 . One need to explain the application scenario and then choose the best scheme with proper justification.
Response 3:Thank you very much for your suggestion, I think for this is a better direction for discussion and we will add it in the discussion of LFRU or TLRU program.
4) Why do both algorithm 1 and 2 has the same title “Content cache placement algorithm based on content popularity”
Response 4:I'm very sorry, I've made changes to this section.
5) Please include a stepwise explanation of the proposed algorithms.
Response 5:I'm very sorry, I've made changes to this section.
6) Authors stated that “The local popularity can then be calculated based on the user’s own characteristics, combined with the historical request records of users associated with the same AP”. Why do we need to find the local popularity based on a user’s local characteristics? It will be more appropriate to consider the location-based popularity by taking the edge devices into consideration.
Response 6:Thank you very much for your suggestion, I rephrased this part.
Reviewer 2 Report (Previous Reviewer 1)
The article is devoted to the management of communication resources in a centralized network infrastructure. The topic of the article is relevant. The structure of the article does not correspond to that adopted in the MDPI. The article does not contain a Discussion section. The level of English is acceptable. Figure 4 is too small. The article cites 24 sources, not all of which are relevant.
The following remarks can be made about the material of the article:
1. To simulate the studied caching process, the authors chose the mathematical apparatus of Markov models. The authors chose their simplest version - the Markov chain. As is known, in such a model, the next state does not depend on the previous states of the simulated process. The authors did not substantiate whether such an approximation is acceptable in the context of the problem under study.
2. The calculation of the parameters of the Markov chain is a rather laborious process. The authors did not describe at all how they calculated the parameters of their model. What algorithms or frameworks were used for this?
3. The most important stage of mathematical modeling is to check the adequacy of the resulting model. This is done using the methods of probability theory and mathematical statistics using appropriate static criteria and table functions. This is not done in the article. Once again, I draw your attention to the fact that the absence of the Discussion section is unacceptable.
4. The authors did not describe the source of the data used to conduct the experiments. This is all the more relevant, because the authors are exploring a rather exotic process.
5. In the Conclusions section, the authors forgot to mention the direction of further research. In general, this article gives me a feeling of deja vu. I definitely saw Figures 1 and 3 in one of the articles I reviewed for Symmetry or Applied sciences this month.
Author Response
Response to Reviewer 2 Comments
We thank the reviewer for comments. Our reply is detailed below.
Revised as suggested.
The article is devoted to the management of communication resources in a centralized network infrastructure. The topic of the article is relevant. The structure of the article does not correspond to that adopted in the MDPI. The article does not contain a Discussion section. The level of English is acceptable.
Figure 4 is too small.
Response :Thank you very much for your suggestion, I modified this one picture.
The article cites 24 sources, not all of which are relevant.
Response : Thank you very much for your suggestion, I have rearranged the references and added some more literature.
The following remarks can be made about the material of the article:
- To simulate the studied caching process, the authors chose the mathematical apparatus of Markov models. The authors chose their simplest version - the Markov chain. As is known, in such a model, the next state does not depend on the previous states of the simulated process. The authors did not substantiate whether such an approximation is acceptable in the context of the problem under study.
Response 1:Thank you very much for your suggestion, I think it is a good one, also we think that in this paper about Markov problem is just to analyze the throughput and is not the focus of this paper.
- The calculation of the parameters of the Markov chain is a rather laborious process. The authors did not describe at all how they calculated the parameters of their model. What algorithms or frameworks were used for this?
Response 2:Thank you very much for your suggestion, and the literature we cite covers this issue in detail.
- The most important stage of mathematical modeling is to check the adequacy of the resulting model. This is done using the methods of probability theory and mathematical statistics using appropriate static criteria and table functions. This is not done in the article. Once again, I draw your attention to the fact that the absence of the Discussion section is unacceptable.
Response 3:Thank you very much for your suggestions, I have added these in the last part
- The authors did not describe the source of the data used to conduct the experiments. This is all the more relevant, because the authors are exploring a rather exotic process.
Response 4:Thank you very much for your suggestions, I have added these in the conclusion section
Special thanks to you for your good comments.

Round 2
Reviewer 1 Report (New Reviewer)
I have no further comments
Reviewer 2 Report (Previous Reviewer 1)
I made the following remarks to the basic version of the article:
1. To simulate the studied caching process, the authors chose the mathematical apparatus of Markov models. The authors chose their simplest version - the Markov chain. As is known, in such a model, the next state does not depend on the previous states of the simulated process. The authors did not substantiate whether such an approximation is acceptable in the context of the problem under study.
2. The calculation of the parameters of the Markov chain is a rather laborious process. The authors did not describe at all how they calculated the parameters of their model. What algorithms or frameworks were used for this?
3. The most important stage of mathematical modeling is to check the adequacy of the resulting model. This is done using the methods of probability theory and mathematical statistics using appropriate static criteria and table functions. This is not done in the article. Once again, I draw your attention to the fact that the absence of the Discussion section is unacceptable.
4. The authors did not describe the source of the data used to conduct the experiments. This is all the more relevant, because the authors are exploring a rather exotic process.
The authors answered all of them. Of course, not all of their answers pleased me enough. However, this does not reduce the overall good impression that this article makes. I support its publication.
This manuscript is a resubmission of an earlier submission. The following is a list of the peer review reports and author responses from that submission.
Round 1
Reviewer 1 Report
The article is devoted to one of the applied tasks of the info-communication technologies industry. The topic of the article is relevant. The structure of the article differs from the classical one for research articles (there is no Discussion section). The level of English is acceptable. The article is easy to read. The figures in the article are small and not informative. The article cites 16 sources, not all of which are relevant.
The following remarks can be made about the material of the article:
- I have been using the mathematical apparatus of Markov chains and queuing systems for more than ten years and for the first time I come across the term "bidirectional Markov model". I ask authors to theoretically substantiate the consistency of such a mathematical construct. My bewilderment at the existence of the term "bidirectionality" only increases in view of the topic of the caching process investigated by the authors (Fig. 1). Sorry, but if the video stream of augmented reality glasses does not go from them to the server, but on the contrary, the glasses will “crack” )
- From the material presented in Section 3, one gets the impression that the authors do not denounce the theory of Markov chains from the theory of graphs. I ask authors to analytically substantiate the difference. What variables in the transition matrix (8) are controllable? What method is used to calculate the metric of qualitative indicators of the process under study? Why was this method chosen?
- In Fig. 5 shows graphs comparing the throughput of various caching schemes. Periodic pulsations are noticeable on the graphs, which contradicts the mechanics of the actual caching process on video hosting. Obviously, renting excess bandwidth is inefficient. It is necessary to analyze the dynamics of the use of the communication channel by the target objects, rent the necessary communication resources and demonstrate the effectiveness of session management in conditions of limited resources.
- It seems that the authors themselves did not understand why they brought Fig. 6. I come to this conclusion because I did not find any comments regarding its content in the article.
- The authors need to statistically substantiate the adequacy of the obtained model of the caching process of video streams (using the corresponding statistical criteria and table values).
Reviewer 2 Report
This work talks about a content-aware proactive VR video caching for cache-enabled AP over edge networks. The authors performed a careful and thorough review of the literature, as the section was very informative and substantial. Appropriate theoretical framework was applied. I found the methodological part to be well justified and reasonable for this type of analysis. Although the manuscript is overall well-written and structured, it might benefit from additional spell/language checking. However, I have some comments which I would like to be addressed before the acceptance of this paper.
- What was the key motivation behind using the classical two-way Markov analysis model?
- Add more explanation about the global and local popularity?
- Explain the simulation parameters in detail.
- Improve the quality of figure 4.
- Line 315, what is meant by user’s saliency?
- What are the limitations of the preset study?